# Factors Influencing College Students’ Mental Health Promotion: The Mediating Effect of Online Mental Health Information Seeking

**DOI:** 10.3390/ijerph17134783

**Published:** 2020-07-03

**Authors:** Wenen Chen, Qian Zheng, Changyong Liang, Yuguang Xie, Dongxiao Gu

**Affiliations:** 1The School of Management, Hefei University of Technology, Hefei 230009, China; xsccwe@hfut.edu.cn (W.C.); cyliang@hfut.edu.cn (C.L.); 13095537911@163.com (Y.X.); gudongxiao@hfut.edu.cn (D.G.); 2The School of Management, Anhui Science and Technology University, Fengyang 233100, China; 3Key Laboratory of Process Optimization and Intelligent Decision-making of Ministry of Education, Hefei 230009, China

**Keywords:** college students’ mental health, information network society, online health information searching, the external quality of Internet platforms, electronic health literacy

## Abstract

For college students, mental health is an important factor in ensuring their ability to study and have a normal life. This research focuses on factors affecting the mental health of college students in the information network society. We constructed a theoretical model that influences their online mental health information seeking behavior from internal and external perspectives, and by extension, affects their mental health. Through the data obtained by field research and questionnaire survey on the online mental health information seeking behavior of some college students in Internet health information platforms, a structural equation model is used to test the hypotheses. Results show that the quality of external Internet platforms and the quality of internal electronic health literacy have a significantly positive impact on the online health information searching behavior of college students; electronic health literacy and online mental health information seeking behavior have significantly direct positive effects on college students’ mental health. Further, online health information searching behavior has a significant mediating effect between Internet platform quality, electronic health literacy, and college students’ mental health. The research conclusions have theoretical value and practical significance to study the factors influencing college students’ mental health in the context of information network society.

## 1. Introduction

College students are one of the backbone forces for the future development of society [1]. According to statistics, there are more than 40 million college students in China alone. For college students living far away from their families, mental health is one of the important factors in ensuring their ability to study and have a normal life. A survey of 2018 white papers on the mental health of China’s urban residents concluded that 73.6% of college students were in a state of mental sub-health. The problems were most acute among those who were highly educated. The mental health problems of college students are mainly manifested in interpersonal communication disorders, Internet addiction, and emotional problems [2,3]. These mental health problems can lead to bad behaviors such as excessive drinking, illegal-drug use, skipping school, irritability, and even suicide, and can seriously affect college students′ ability to study and their quality of life. One could argue that the state of college students’ mental health is one of the most important issues that society should pay close attention to.

College students are the “natives” of the information society, born and raised in the age of Internet information; the breadth, intensity, and duration of their online searching surpasses that of other groups [4], and the ability to perceive and use data is also higher than other groups [5]. Information about mental health is an important part of the material searched on the Internet [6]. With the help of Internet platforms, college students usually choose to search for information about mental health on relevant medical websites and app; or by communicating with online doctors to determine whether their mental state is at a healthy level. From the internal perspective of information search subjects, Canadian scholar C.D. Norman and others hold that the influencing factors of online information searching have a positive correlation with electronic health literacy at the perspective of individual ability [7]. According to Delone and McLean’s successful model of information systems, people’s online behavior is motivated mainly by the quality of external information [8]. The emergence of individual behavior is the result of the comprehensive effect of external environmental factors and internal subjective factors, so it is necessary to consider them together. In addition, the ultimate goal of college students’ online health information searching is their mental health. Shaw and Gant think the Internet can positively promote the physical health of individuals [9]; Aref-Adib and other scholars believe that online mental health information seeking behavior (OMHISB) has a positive relationship with mental health [10]; Peris and others think that good online relationships with other people have a positive effect on mental health [11]. In this information network society, factors affecting the mental health of college students need to be further explored. Therefore, this study deals with college students’ mental health, focusing on the following three issues:What are the internal and external factors that influence college students’ OMHISB?What is the comprehensive effect of online health mental information seeking behavior and college students’ electronic health literacy on mental health promotion?What is the role of OMHISB in the relationship between internal and external factors and mental health promotion?

The remaining sections are arranged as follows: Section 2 introduces the main theoretical background and research hypotheses of the study; Section 3 makes an empirical analysis based on survey data; Section 4 expounds on the empirical analysis results; Section 5 discusses the theoretical and practical significance of the study; and Section 6 is the research conclusion.

## 2. Theoretical Background and Research Hypotheses

### 2.1. Theoretical Background

#### 2.1.1. Social Cognitive Theory

Social cognitive theory was first proposed by Bandura, an American psychologist [12], and has become one of the important theories in the fields of pedagogy and social psychology. The core of social cognitive theory is the theory of ternary interaction, that is, the interaction among individual cognition, environment, and behavior. It challenges the traditional view that individual behavior is determined only by internal factors or the external environment. A person’s behavior will be affected not only by their own cognition, but also by changes in the external environment. Of course, when individual behavior changes, it will also affect the person’s own cognition [12]. Moreover, social cognition theory, which has been verified by many previous studies, explains individual behavior variables well [13]. In this paper, the electronic health literacy of college students is regarded as the internal individual’s self-awareness, and the quality of Internet platform is regarded as the external environment variable that affects college students’ OMHISB. The influence mechanism model of college students’ mental health is constructed from the internal and external perspectives.

#### 2.1.2. Information Systems Success Model

The success model of information systems was proposed by Delone and McLean in 2003 on the basis of their research ten years earlier [8]; it is considered an important point of departure for evaluating the quality of information systems/platforms [14]. Furthermore, Delone and McLean divided it into three dimensions: information quality, system quality, and service quality [8,15]. Information quality is the key variable used to measure semantic success, which reflects the degree of information integrity, accuracy, and real-time output in the form of reports or online screens during the use of the system/platform [16]. System quality is the key variable used to measure technical success, reflecting the reliability and efficiency of the system/platform in processing its own software and data [17]. Service quality is also a key variable by which to measure semantic success [18,19], reflecting the credibility, timeliness, personalization, and specialization of system/platform service quality [17]. Moreover, many previous studies have verified the validity of the model, showing that these three dimensions are one of the key factors in determining the success of an information system/platform [15], and will also have a certain impact on individual behavior. This paper uses the three-dimensional division of Delone and McLean (i.e., information quality, system quality, and service quality) to measure the quality of Internet platforms currently used by college students, and then conducts in-depth research to explore its effect on college students’ OMHISB.

Before conducting OMHISB, many college students may form the expected quality of online medical platform [20] through other’s word-of-mouth on online medical platforms, or their previous online medical search behavior experience for non-mental health problems. As a result, it will produce students’ intuitive experiences of the system, information, and service quality of the online medical platform. Then, when they face some mental health problems, they will consider whether to conduct a mental health consultation online. Therefore, this paper draws on previous studies and incorporates expected quality into the model for further analysis [21]. 

#### 2.1.3. Online Mental Health Information Seeking Behavior

Health information seeking behavior is a concept derived from information seeking, which is defined by Niederdeppe as the behavior that aims to obtain health information in response to specific related events [22]. However, with the development of electronic health care, the Internet has become an important way for people to obtain health information. Some scholars are gradually turning their attention to in-depth research on OMHISB. Among them, Yanxia and Xinxin summarized previous research and defined the concept of OMHISB as the behavioral process in which individuals, guided by their own health information needs, use Internet tools to search, browse, evaluate, select, and use health-related knowledge or information to meet those needs [23]. Subsequently, OMHISB has been studied by more in-depth and developed research, which is of great significance to the current social background of booming Internet use. College students use the Internet more frequently than any other group [4]. How to effectively use college students’ OMHISB to promote the development of their mental health has important research value.

#### 2.1.4. Mental Health

The study of mental health appeared early, but the initial connotation of mental health focused more on the negative aspects of individuals [24] (i.e., the stage of negative mental health [25]). However, with the continuous development of positive psychology theory, many scholars have found that focusing only on negative aspects makes the study of individual mental health one-sided, and thus the positive aspects of mental health have gradually become the focus of scholarly research [26]. The dual factor model of mental health is established and supported effectively [15] and consists of two dimensions: positive mental health and negative mental health. Negative mental health refers to the elimination of individual mental illness, whereas positive mental health refers to individual happiness and a positive mental state [27]. Based on the dual factor model of mental health, the topic of college students’ mental health explored in this paper includes both the elimination of individual mental illness and the degree of happiness they perceive.

### 2.2. Research Hypotheses

#### 2.2.1. Internet Platform Quality and Online Mental Health Information Seeking Behavior

The success model of an information system indicates that information quality, system quality, and the service quality of the system/platform can on the one hand affect user satisfaction, and on the other hand affect users’ intention and behavior [8]. However, when looking at research on the success model of information systems, previous researchers focused on the application of a certain information system or specific platform, and then judged whether the construction of this information system or platform was successful [15]. With the development of Internet technology, Internet platforms gradually began to be integrated into all aspects of life, and in the process of using them, people’s perceptions of platform quality can effectively affect their behavior and willingness to use [28,29]. Saeed and others studied users of online retailers, and pointed out that the information quality, system quality, and service quality of online retail platforms have a significant effect on online consumer behavior [30]; similarly, Ho et al. researched online searching behavior of users and found that the quality of websites—information quality, system quality, and service quality—can affect user online information searching [31]. In addition, as a group, college students use the Internet more frequently than the general population [4]; their perception of Internet platform quality also has an impact on their OMHISB. Based on the above research, the following assumptions are put forward on the external perspective:

**Hypothesis** **1** **(H1).**
*The information quality of Internet platforms has a significantly positive impact on the online mental health information seeking behavior of college students.*


**Hypothesis** **2** **(H2).**
*The system quality of Internet platforms has a significantly positive impact on the online mental health information seeking behavior of college students.*


**Hypothesis** **3** **(H3).**
*The service quality of Internet platforms has a significantly positive impact on the online mental health information seeking behavior of college students.*


#### 2.2.2. Electronic Health Literacy and Online Mental Health Information Seeking Behavior

Health literacy is an important variable of individual health perception, first proposed by S.K. Simonds in 1974. Health literacy refers to people’s ability to acquire and understand health information and use this information to improve their health [32]. However, the development of Internet technology has gradually allowed the Internet to become one of the most important resources for individuals to obtain health information [6]. Canadian scholars, Norman and Skinner, put forward the concept of electronic health literacy for the first time with consideration of the characteristics of contemporary society to refer to the multidimensional abilities to obtain medical health information from electronic resources and use the obtained health information to solve one’s own health problems [7]. Subsequently, many scholars have carried out in-depth research on this concept and found that electronic health literacy has an important impact on individual OMHISB. Previous studies have shown that individuals with low levels of health literacy tend to use fewer online portals to search for health information [33]; Britt and Hatten conducted research on college students and found that individuals with high levels of electronic health literacy are more likely to search for health information on the Internet [34]. Jensen found in his research that individuals with low health literacy use less Internet technology to improve their health [35]. Hanik and Stellefson also conducted research on college students and discovered that improving their electronic health literacy has an important impact on their OMHISB [36]. Based on the above, the following assumption is proposed with respect to the internal perspective of individuals:

**Hypothesis** **4** **(H4).**
*College students’ electronic health literacy has a significantly positive impact on their online mental health information seeking behavior.*


#### 2.2.3. Online Mental Health Information Seeking Behavior, Electronic Health Literacy, and Mental Health

The continuous development of the Internet and mobile technology has changed traditional ways of obtaining and understanding health information [10]. From the internal point of view, when individuals search more health information through the Internet, they can effectively understand their own health status, and reduce anxiety, suspicion, and other negative emotions, thereby improving their mental health [37]. Similarly, when people’s levels of electronic health literacy are higher, they are more sensitive to health information and pay more attention to their own health status, thereby improving their mental health. Aref Adib et al. found that patients’ OMHISB can effectively improve their mental health status; they also discussed the acceptability of mobile mental health applications [10]. Based on the above, the following assumptions are proposed:

**Hypothesis** **5** **(H5).**
*Electronic health literacy has a significantly positive effect on college students’ mental health.*


**Hypothesis** **6** **(H6).**
*Online mental health information seeking behavior has a significantly positive effect on college students’ mental health.*


#### 2.2.4. The Mediating Role of Online Mental Health Information Seeking Behavior

Previous studies on the mediating effects of OMHISB are inadequate. Most focused on the previous variables, namely the influencing factors of OMHISB. Externally, the individual’s perception of the quality of Internet platforms rarely directly affects their mental health, but the search of Internet health information is the key to connecting the two variables. McKinley and Wright pointed out in their research that individuals’ perceptions of information support can trigger OMHISB, leading to a healthier lifestyle [38]. This shows that individuals’ perceptions of the external environment can effectively influence physical and mental health with the mediating role of individuals’ behaviors in promoting their health. Internally, on the one hand, college students’ levels of electronic health literacy will directly affect their mental health; on the other hand, it will further affect the mental health through OMHISB. Dunne et al. found that health promoting behaviors play a mediating role between self-sympathy and physical health [39]. Based on the above, the following assumptions are proposed:

**Hypothesis** **7a** **(H7a).**
*Online mental health information seeking behavior plays a mediating role between information quality and college students’ mental health.*


**Hypothesis** **7b** **(H7b).**
*Online mental health information seeking behavior plays a mediating role between system quality and college students’ mental health.*


**Hypothesis** **7c** **(H7c).**
*Online mental health information seeking behavior plays a mediating role between service quality and college students’ mental health.*


**Hypothesis** **7d** **(H7d).**
*Online mental health information seeking behavior plays a mediating role between electronic health literacy and college students’ mental health.*


Based on the above research hypotheses and the social cognitive theory and the success model of information system, this paper brings the qualities of the Internet platform—information quality, system quality, and service quality—into the model from the external side, and electronic health literacy into the model from the internal side. Through the mediating role of OMHISB, the influence mechanism model of college students’ mental health is constructed, as shown in Figure 1. 

## 3. Research Methodology

### 3.1. Measurement

The main potential variables of this research model are information quality (IQ), system quality (SYQ), service quality (SEQ), electronic health literacy (EHL), online mental health information seeking behavior (OMHISB), and mental health (MH). The measurement indexes of seven potential variables are shown in Table 1. The questionnaire is in the form of a Likert 7-point scale. From 1 to 7, the numbers represent the degree of the respondents’ agreement with the question; 1 represents complete disagreement, and 7 represents complete agreement.

The qualities of Internet platforms—information quality, system quality, and service quality—are based on the research of Wixom and Todd, Zha, and others [18,40], and are modified according to the actual situation. Finally, the questionnaire required in this paper was formed with a total of 11 measurement items. There are three items of information quality, five items of system quality, and three items of service quality. For example, “I think the health information in the Internet medical platform is updated in time”; “I think the Internet medical platform is reliable”; and “I think the Internet medical platform provides reliable services”.

Electronic health literacy is mainly based on the scale compiled by Norman and Skinner [7], and is modified according to the actual situation. There are four measurement items in total. For example, “I know where to find useful health resources on the Internet.”

Online mental health information seeking behavior, based on the research questionnaire compiled by Zha et al. and Yan and Davison, has three measurement items [18,42]. For example, “I often look for health information on the Internet medical platform.” 

Mental health is based on the hospital anxiety and depression scale compiled by Zigmond and Snaith [43,44]. In this paper, combined with the group characteristics of college students and the actual investigation, we make appropriate modifications to a total of five measurement items. For example, “I feel happy” and “I look forward optimistically to everything.” 

### 3.2. Respondents and Data Collection

To verify the rationality and validity of the theoretical model proposed in this paper, this study uses a questionnaire survey method to collect firsthand data for empirical analysis. By 2020, there were more than 2600 institutions of higher learning and more than 40 million college students in China. In this study, by means of Internet and telephone interviews, we conducted a questionnaire survey on the OMHISB of some college students in Internet health information platforms.

This study uses the method of random questionnaire to investigate college students in China. First of all, to verify the validity of the questionnaire, we first conducted field research and distributed the questionnaire in a university in Anhui Province. A total of 50 copies was distributed in the early stage for pre-investigation in a college in Anhui, and items were modified based on the results. Subsequently, the questionnaire was revised and distributed in colleges and universities nationwide. In addition, in the questionnaire survey, in order to ensure that the subjects conducted an online search for mental health issues, in the questionnaire, we clearly pointed out that the attention of the subjects is an investigation of the online search behavior and effect of mental health problems, and through the reverse question to ensure that the subject was of online mental health search behavior. In the later questionnaire sorting, questionnaires without OMHISB were eliminated. In this survey, a total of 550 questionnaires was issued. After eliminating the invalid questionnaires, 491 valid questionnaires were collected, for a total effective rate of 89.27%. The specific sample data statistics are shown in Table 2.

## 4. Results

### 4.1. Measurement Model Evaluation

The partial least squares method or maximum likelihood method were used to analyze the structural model. The partial least squares method is widely used in exploratory factor analysis because of its relatively loose requirements for normal distribution of sample data quality and its flexibility in dealing with missing data. In addition, there was no significant difference between the two methods in the analysis of large sample data. Therefore, this paper mainly used SPSS20 (IBM, Armonk, NY, USA) and SmartPLS3.0(SmartPLS GmbH, Bönningstedt, Germany) for analysis, and partial least squares regression structural equation model (SEM) to verify the measurement model and structural model.

#### 4.1.1. Descriptive Statistical Analysis

First, a descriptive statistical analysis of sample data was carried out, and the analysis results are shown in Table 3. According to the analysis results, the mean value of each variable item of the model ranged from 4.326 to 4.89, and the standard deviation range was from 1.484 to 1.738, which shows that the data are relatively centralized, the fluctuation is small, and the model has good adaptability. In addition, the factor load of each item ranged from 0.834 to 0.912, which is higher than the threshold value of 0.7. This shows that the measurement index of each latent variable has high correlation and convergence validity [45], indicating that the items set in the questionnaire are reasonable.

#### 4.1.2. Common Method Bias Assessment

Considering the questionnaire survey, the data was collected from one single source, which may lead to common method bias. This paper adopted program control methods, including two methods of process control and statistical control, to ensure that the questionnaire did not produce serious common method bias problems. Process control was mainly used for anonymous filling of the survey to reduce the ambiguity of the questionnaire and improve the clarity of the questionnaire, as far as possible to avoid errors caused by factors such as inaccurate concept measurement and mutual interference between responders. In addition, this paper also adopted a statistical control method, that is, Harman’s single factor test was performed on the recovered data, and the principal component analysis method was used to perform factor analysis on the total variance of the interpretation of the measured indicators. The results show that the total measurement index has a total of six principal components (eigenvalues > 1), and the variance of the first principal component factor is 30.01%, which is lower than the threshold of 40%, showing that a single principal component does not have very strong explanatory power. It can be seen that the sample data survey results of this paper do not have serious common method bias problems, and the explanatory power of all measurement index items is within an acceptable range.

#### 4.1.3. Reliability and Validity Test

The measurement quality of the questionnaire can be analyzed for its reliability and validity. The test results are shown in Table 4.

From the reliability test of the questionnaire, Cronbach’s α value is usually used for observation. Cronbach’s α value reflects the reliability of the measurement indicators of the potential variables in the questionnaire. The higher the value, the higher the reliability of the scale. In this model, Cronbach’s α of all measurement variables is greater than the recommended value of 0.7 [45], and its range is from 0.863 to 0.917, which shows that the reliability of the questionnaire is good.

Based on the test, the validity of the questionnaire can be divided into aggregation validity and differentiation validity. First, let us adapt the perspective of aggregation validity. Aggregation validity is a test to measure the consistency of multiple measurement items under the same construct. Hair et al. pointed out that factor load, composite reliability (CR), and average variance extraction (AVE) are three effective indicators to test the polymerization validity [46]. The factor load of the model is shown in Table 3, and the factor load of each observation variable exceeds the recommended value of 0.6, indicating that the observation variable of the model has a high correlation with the structural variable [47]. The CR value describes the degree to which observation variables determine the potential structure, and also reflects the degree of internal consistency of variables. The CR value of the model ranges from 0.916 to 0.938, which is 0.7 higher than the recommended value, indicating that the internal consistency of the model variables is good, and the observation variables set can better explain the potential model structure. The AVE value reflects the total difference of indicators occupied by potential structure. The AVE in this model ranges from 0.721 to 0.817, beyond the recommended value of 0.5 [46]. It shows that the observation variables in this model explain each measurement dimension well [48].

Discriminative validity refers to the degree of uncorrelation between the measurement indexes of each latent variable and the measurement indexes of other latent variables [49]. The discrimination validity can be measured by the square root of each AVE observation variable. In this model, the square root of each AVE observation variable is greater than the correlation coefficient between each AVE observation variable and other observation variables, indicating that each observation variable has a strong discrimination coefficient and a high degree of discrimination.

According to the above analysis, the questionnaire design of this study is reasonable and has good reliability and validity. The design of the measurement model is effective and reasonable, which can be used for further structural model fitting analysis.

### 4.2. Structural Model Analysis

#### 4.2.1. Total Effect Analysis

The structural model points out the causal relationship among the variables in the model [50]. In this study, the theoretical model proposed by SmartPLS3.0 software is used for structural equation analysis, and the analysis results are shown in Figure 2.

From the analysis results of the above model, we can draw the following conclusions: First, it analyzes the influence of internal and external factors on college students’ OMHISB. Among the factors, the information quality of external Internet platform (*β* = 0.160, *p* < 0.01), system quality (*β* = 0.275, *p* < 0.01), and service quality (*β* = 0.312, *p* < 0.01) have significantly positive effects on college students’ OMHISB; the internal perspective of individual electronic health literacy (*β* = 0.164, *p* < 0.01) also has significantly positive effects on college students’ OMHISB, and the common explanation variance is 58.6%. Moreover, from the above analysis results, it can be seen that for college students, the effect of external factors is greater than that of internal factors, in that the impact of Internet platform quality on college students’ OMHISB is greater than their own awareness and initiative for health. It is assumed that H1, H2, H3, and H4 are all verified.

Secondly, it analyzes the direct effect of improving the mental health of college students. From the above analysis results, we can see that the individual’s electronic health literacy (*β* = 0.271, *p* < 0.01) and OMHISB (*β* = 0.299, *p* < 0.01) have a significantly positive impact on the improvement of college students’ mental health, with a total explanatory variance of 70.3%, justifying the variables to a large degree. Among them, OMHISB has the greatest impact on their mental health, followed by electronic health literacy. It is assumed that H5 and H6 are all verified. The mediating role of OMHISB still needs further analysis.

Thirdly, it analyzes the effect of mental health of college students after adding control variables. *t*-Test results show that gender, education level and, whether there is clear historical mental disease diagnosis have no significant difference on college students’ mental health. This means that attribute variables have no significant effect on empirical analysis results. 

#### 4.2.2. Mediating Effect Analysis

To test the mediating effect of OMHISB, this study established two models and used SmartPLS3.0 software for structural model analysis. Model 1 includes only variables such as Internet platform quality, electronic health literacy, and mental health, and analyzes the impact of direct effects. The analysis results are shown in Figure 3. Model 2 adds mediating variables, namely OMHISB, to test its mediating role on the basis of Model 1. The analysis results are shown in Figure 4.

In Model 1, the quality of Internet platform and electronic health literacy has some significant effects on the improvement of college students’ mental health. Among them, the system quality (*β* = 0.269, *p* < 0.01), service quality (*β* = 0.325, *p* < 0.01), and individual electronic health literacy (*β* = 0.284, *p* < 0.01) have a significantly positive impact on the improvement of college students’ mental health, and the mediating effect of OMHISB can be considered [51]. However, information quality (*β* = 0.063, *p* > 0.01) has no significant effect on the improvement of mental health, so it is impossible to consider the mediating role of OMHISB, assuming H8a is not tenable.

In Model 2, compared with Model 1, the regression coefficient between the system quality of Internet platforms and the mental health of college students decreased from *β* = 0.269 to *β* = 0.194 and the *t*-value decreased from 5.723 to 4.011 after the addition of the OMHISB variable. This shows that OMHISB plays a mediating role between the system quality and the mental health of college students. H8b is verified. Similarly, OMHISB plays a mediating role in service quality, electronic health literacy, and the mental health of college students, assuming that H8c and H8d are verified. Therefore, Internet health information seeking plays a mediating role between the quality of an Internet platform and electronic health literacy, and the mental health of college students, assuming that H7 is partially established. The final hypothesis test results are shown in Table 5.

## 5. Discussion

### 5.1. Implications for Research

Based on the internal and external perspectives, this paper explores the influence mechanism of college students’ mental health through the mediating role of OMHISB, which mainly has the following theoretical contributions:

First, through empirical research methods, this paper puts forward the influence mechanism model of college students’ mental health and proves it. Previous scholars have studied more directly the existing mature mental health scale and directly through the scale score to analyze their mental health status. The research on the impact mechanism of their mental health is insufficient and needs to be improved [15]. In addition, there are a wide range of studies on the impact mechanism of college students’ mental health [51,52]. However, few scholars analyze the impact mechanism of college students’ mental health from the perspective of college students’ information technology and Internet application. Considering that college students are a community that frequently uses the Internet [4], the Internet also has an important impact on their mental health. This paper explores the influence mechanism of college students’ mental health in terms of internal electronic health literacy and external Internet platform quality, enriches the theoretical basis of the influence mechanism of college students’ mental health, and provides the theoretical basis for follow-up research.

Second, in this paper, OMHISB as a variable is included in the theoretical model to analyze the impact of its mediating effect. Previous studies on the mediating effect test of OMHISB are inadequate, while more research focuses on its previous variables, namely the influencing factors of OMHISB [15]. This paper includes OMHISB in the model and verifies the mediating effect of internal and external factors on college students’ mental health, which enriches the theoretical basis of OMHISB. In addition, this study finds that the mediating effect of OMHISB between information quality and the mental health of college students cannot be established, which is different from previous research [53]. Considering the differences of research targets, since health information is professional information in general, it is difficult to accurately judge the quality of health information for ordinary nonmedical college students [54]. Therefore, there is no significant mediating correlation between information quality and mental health in Internet health information searching behavior. It suggests that we should pay attention to the differences of research situations in studying the factors and behavioral results of Internet behavior.

Third, in this paper, we built a model of the impact mechanism of college students’ mental health from both external and internal perspectives. The external perspective combines with the development of information technology to consider the quality of the network platform, and the internal perspective combines the individual’s acceptance of information technology to consider the internal electronic health literacy of college students, which lead to a more comprehensive analysis of the impact mechanism of college students’ mental health. The previous research is more from a single perspective [38], or to compare the impact mechanism of college students’ mental health in different research background [55]. This study analyzes in-depth on the basis of the previous one, which also enriches the theoretical basis of college students’ mental health and provides an important theoretical basis for subsequent research.

### 5.2. Implications for Practice

College students’ mental health has always been the focus of social concern, and certain cultural or social changes have made maintaining mental health more challenging. Their mental health problems may lead to bad behaviors that affect their ability to study and their quality of life, such as alcohol, irritability, and even suicide. This paper studies and analyzes the mental health of college students from the perspective of Internet use, which is of great practical significance in terms of alleviating their mental health problems to a certain extent.

Firstly, from the individual point of view, using the Internet to search for relevant health information and giving full play to the advantages of the Internet can help college students effectively understand their health status and reduce anxiety, suspicion, and other negative emotions [38] so as to improve their mental health. This has important practical significance.

Secondly, from the perspective of platform managers, improving the quality of the health care platform, including information quality, system quality, and service quality, is of great value. Improving the quality of the platform by optimizing its design and strengthening data management will help promote college students’ searching for health information on the Internet, and it has an important role in improving their mental health.

Thirdly, colleges and universities have an incentive to improve students’ electronic health literacy, an essential component of institutions of higher education, because it can effectively improve their mental health.

### 5.3. Limitations and Future Research Directions

Based on the internal and external perspectives, this paper explores the influence mechanism of college students’ mental health through the mediating role of OMHISB. This has certain theoretical value and practical significance, but there are also some shortcomings. First, this paper is about college students in China, which has certain geographical limitations and research object limitations. Second, Internet platforms mentioned in this paper generally refer to online medical platforms used by various college students. The quality of different platforms is uneven, yet this paper does not distinguish between them. This needs to be further improved in the future.

## 6. Conclusions

This study constructs a theoretical model that influences college students’ OMHISB from internal and external perspectives and by extension affects their mental health. Through field research and questionnaires, a structural equation model is used to test the hypotheses. The main results are as follows.

Firstly, the quality of external Internet platforms and the quality of internal electronic health literacy have a significantly positive impact on the online health information searching behavior of college students. The results show that the three dimensions of Internet platform quality—information quality, system quality, and service quality—can effectively promote college students’ OMHISB, and can improve college students’ consciousness of actively searching for health information on Internet platforms [29,30]. Moreover, students’ perception of platform service quality has the greatest impact on their online health searching, whereas the platform’s information quality and the internal perspective of electronic health literacy have relatively little impact. This shows that with changes over time, contemporary college students’ perception of the quality of information while searching the Internet for health information is no longer the main focus of attention; rather, the quality of service occupies an important position. More attention is paid to the services received in the process of searching for information, which is a breakthrough compared with previous research [31,32]. The possible reason is that health information is generally professional knowledge. Nonmedical college students have difficulty accurately judging the quality of health information [55], so the sensitivity to service quality will be greater than the sensitivity to information quality.

Secondly, electronic health literacy and OMHISB have significantly positive effects on college students’ mental health. The results show that when college students choose to use OMHISB, they will have more opportunities to connect with knowledge about mental health, and they will also have easier access to the required information about mental health. In turn, timely assessment and adjustment of their own mental health status, through the reduction of anxiety, suspicion, and other negative emotions will thereby improve their mental health [38]. Similarly, when their level of electronic health literacy is high, they will pay more attention to their own mental health status, again improving their mental health. 

Thirdly, online health information searching behavior is established in the mediating role of Internet platform quality, electronic health literacy, and college students’ mental health. The results show that Internet health information searching plays a mediating role in the system quality, service quality, electronic health literacy, and mental health of college students. After adding OMHISB variables, the original direct effect is significantly reduced. However, its mediating role between information quality and mental health is not established, and college students’ perception of information quality on Internet platforms will not directly promote their mental health; therefore the mediating role cannot be considered [51]. This may be related to the influence of the change in contemporary college students’ values, and more attention should be paid to the quality of service in information searches. At the same time, it may also be caused by the high degree of professionalism of health information, which makes it difficult for college students to form an effective perception of its quality [55].

## Figures and Tables

**Figure 1 ijerph-17-04783-f001:**
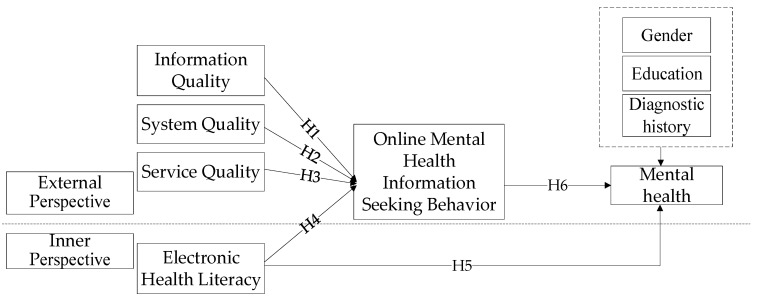
Research model.

**Figure 2 ijerph-17-04783-f002:**
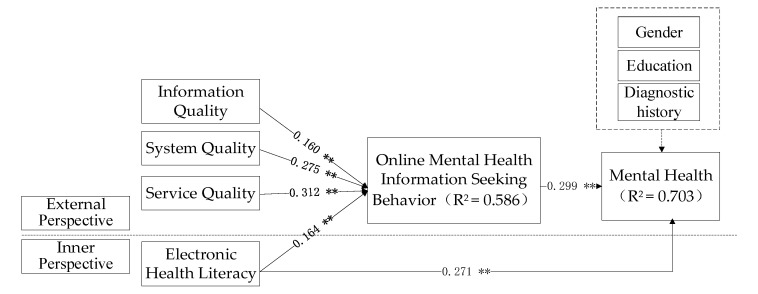
Model fitting diagram. Note: ** means *p* < 0.01.

**Figure 3 ijerph-17-04783-f003:**
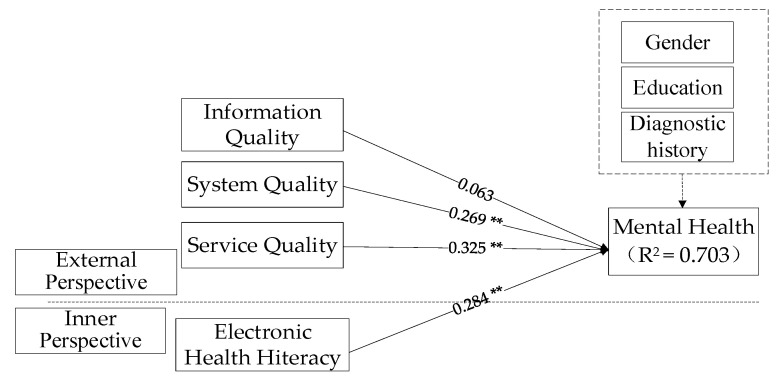
Fitting diagram of direct effect model (Model 1). Note: ** means *p* < 0.01.

**Figure 4 ijerph-17-04783-f004:**
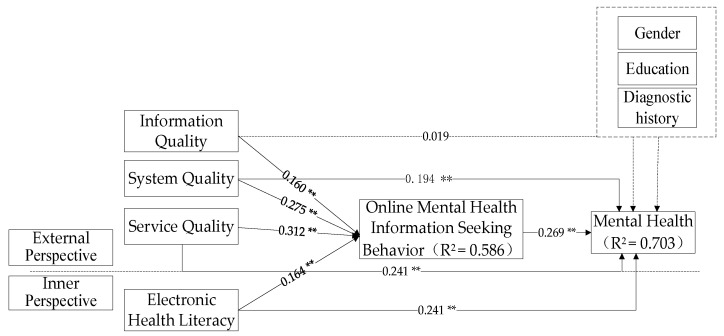
Analysis diagram of mediating effect (Model 2). Note: ** means *p* < 0.01.

**Table 1 ijerph-17-04783-t001:** Variables and indicators used in the research.

Latent Variable	Measurement Item	Reference
Information quality (IQ)	IQ1: The health information in the Internet medical platform is updated in time.	McKinley and Wright, 2014; Dunne et al., 2018 [38,39]
	IQ2: The health information in the Internet medical platform is accurate.	
	IQ3: The health information in the Internet medical platform is comprehensive.	
System quality (SYQ)	SYQ1: The Internet medical platform is reliable.	Shaw and Gant, 2004; Wixom et al. [9,40]
	SYQ2: The Internet medical platform is efficient in searching health information.	
	SYQ3: The Internet medical platform navigation is effective.	
	SYQ4: The layout of the Internet medical platform is clear.	
	SYQ5 The operation of the Internet medical platform is safe.	
Service quality (SEQ)	SEQ1: The Internet medical platform provides reliable services.	Zha et al., 2014; Wixom et al. [18,40]
	SEQ2: The Internet medical platform provides timely services.	
	SEQ3: The Internet medical platform provides personalized services.	
Electronic health literacy (EHL)	EHL1: I know where to find useful health resources on the Internet.	Norman and Skinner, 2006; Shuaijun et al., 2013 [7,41]
	EHL2: I know how to find useful health resources on the Internet.	
	EHL3: I know how to use the obtained Internet health information to help myself.	
	EHL4: I can distinguish between high-quality and low-quality Internet health resources.	
Online mental health information seeking behavior (OMHISB)	OMHISB1: I often look for health information on the Internet medical platform.	Zha et al., 2014; Yan and Davison, 2013 [18,42]
	OMHISB2: I frequently seek health information on the Internet medical platform.	
	OMHISB3: I spend a lot of time looking for health information on the Internet medical platform.	
Mental health (MH)	MH1: I am still interested in the past.	Zigmond and Snaith, 2007; Nunnally, 1978 [43,44]
	MH2: I can laugh and see the good side of things.	
	MH3: I feel happy.	
	MH4: I am optimistic about everything.	
	MH5: I can adjust my mood quickly.	

**Table 2 ijerph-17-04783-t002:** Sample statistics.

College Students	Type	Number	Proportion
**Gender**	Male	235	47.86%
	Female	256	52.14%
**Age**	Under 18 years old	30	6.11%
	18–20 years old	212	43.18%
	20–24 years old	196	39.92%
	Over 24 years old	53	10.79%
**Education**	Junior college	94	19.14%
	Undergraduate	257	52.34%
	Master	88	17.92%
	Doctor and above	52	10.59%

**Table 3 ijerph-17-04783-t003:** Descriptive statistical results of data.

Latent Variable	Variable Item	Mean Value	Standard Deviation	Factor Load
Information quality (IQ)	IQ1	4.766	1.617	0.874
	IQ2	4.89	1.623	0.885
	IQ3	4.868	1.738	0.898
System quality (SYQ)	SYQ1	4.511	1.518	0.859
	SYQ2	4.517	1.672	0.882
	SYQ3	4.53	1.634	0.867
	SYQ4	4.426	1.532	0.859
	SYQ5	4.464	1.565	0.844
Service quality (SEQ)	SEQ1	4.544	1.561	0.9
	SEQ2	4.462	1.651	0.912
	SEQ3	4.642	1.648	0.9
Electronic health literacy (EHL)	EHL1	4.572	1.655	0.875
	EHL2	4.326	1.609	0.873
	EHL3	4.354	1.511	0.869
	EHL4	4.546	1.533	0.871
Online mental health information seeking behavior (OMHISB)	OMHISB1	4.493	1.573	0.878
	OMHISB2	4.546	1.586	0.889
	OMHISB3	4.458	1.557	0.889
Mental health (MH)	MH1	4.426	1.495	0.858
	MH2	4.58	1.522	0.834
	MH3	4.544	1.484	0.844
	MH4	4.77	1.523	0.866
	MH5	4.623	1.546	0.842

**Table 4 ijerph-17-04783-t004:** Reliability and validity test.

Variables	Alpha	CR	AVE	IQ	MH	SEQ	EHL	SYQ	OMHISB
Information quality	0.863	0.916	0.784	0.886					
Mental health	0.903	0.928	0.721	0.503	0.849				
Service quality	0.888	0.93	0.817	0.462	0.719	0.904			
Electronic health literacy	0.895	0.927	0.761	0.562	0.724	0.659	0.872		
System quality	0.914	0.935	0.744	0.493	0.714	0.66	0.706	0.862	
Online mental health information seeking behavior	0.863	0.916	0.785	0.532	0.73	0.676	0.654	0.676	0.886

Abbreviations: CR = composite reliability; AVE = average variance extraction.

**Table 5 ijerph-17-04783-t005:** The estimation results of structural parameters.

Hypothetical Path	Path Coefficient	T Statistics	Conclusion
H1: Information Quality → OMHISB	0.16 **	4.237	Support
H2: System Quality → OMHISB	0.275 **	5.763	Support
H3: Service Quality → OMHISB	0.312 **	7.368	Support
H4: Electronic Health Literacy → OMHISB	0.164 **	3.573	Support
H5: Electronic Health Literacy → Mental Health	0.271 **	8.003	Support
H6: OMHISB → Mental Health	0.299 **	8.547	Support
H7: Mediating Role of OMHISB	-	-	Partial support
Gender → Mental Health	0.011	0.765	No support
Education → Mental Health	0.034	0.891	No support
Diagnostic History → Mental Health	0.029	0.237	No support

Note: ** means *p* < 0.01.

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
