# Peer review of "Factors Influencing College Students’ Mental Health Promotion: The Mediating Effect of Online Mental Health Information Seeking"

_ijerph, 2020, doi:10.3390/ijerph17134783_

Round 1

Reviewer 1 Report

This study focuses on an interesting research question about the impacts of online health information seeking on students’ mental health. Based on the system success model, the authors build up a research model on health information seeking behavior. The theoretical development is rigorous and the research method is appropriate. Overall, the research findings are interesting and have important implications to the literature and practice.

Below are a few comments for further improving the paper.

First, in the introduction, you can give some examples about how college students usually seek mental health information online.

Second, in the abstract, you mention the field study approach in data collection. How is it reflected in the article? Please elaborate or revise it.

Last, you can add a few more references that are published in recent five years.

I find the study well-conducted and well-written. Hope the comments are helpful.

Reviewer 2 Report

I appreciate the opportunity to review the paper. In fact, I took the assignment with great interest after browsing through the abstract. I believe the authors have picked a very interesting and timely research topic, college students’ mental health, both concepts are highly relevant to practices but less understood. Overall the paper is well written, and the research content in paper is substantial and reflects academic rigor required for a premier level outlet. In particular, I like that online health information seeking behavior is selected as the mediator, which provides a fine-grained view to examine the phenomenon. I also like that the authors integrated both internal and external factors to give a holistic understanding of college students’ mental health. In general, the research is interesting and has the potential to contribute to the literature. However, I think that the following issues must be addressed before publishing.

First, I suggest the authors drop the moderating role online social support in the research model. The focus of this paper is about the mediating effects of online health information seeking behavior, which is sufficient for the contribution this paper is expected to make.

Second, the theoretical contribution. If the moderating role online social support was dropped, I suggest the authors add the integration of both the internal and external factors for a holistic understanding of college students’ mental health.

Third, the external level and internal level are confusing. Actually, this paper is not a multi-level study. Please change the level to factor or perspective.

Forth, common method bias. The data was collected from one single source, which may lead to common method bias. Some checks should be conducted to test whether common method bias is a serious problem.

Overall, I enjoy reviewing the paper. I hope the authors will find my comments useful. Best luck with your revision.

Reviewer 3 Report

This is a very detailed analysis of factors affecting the mental health of college students in the information network society by constructing a theoretical model of contributing internal and external factors related to online health seeking behavior and then testing the model. The authors provide a high level of detail to describe the theoretical model, analysis, and results. The statistical methods employed seem fitting. However, I have concerns about the conceptualization and measurement used in the study and in turn, the conclusions that are made. I also feel the paper is unnecessarily long. A shorter paper describing the construction of the theoretical model could maybe be considered, but the empirical analysis and conclusions need to be reframed based on how the data were measured and conceptualized.

I agree that we need further research to understand how to effectively use college students' online health information seeking behavior to promote the development of their mental health. I also agree that online social support is a mechanism used by many college students and could also impact their mental health. Until this paper, I was unfamiliar with the success model of information systems but I see how it has merit in assessing individual internet platforms. I appreciated the discussion in section 2.2.1 that collectively, as internet platforms have become integrated into all aspects of life, perceptions of quality can influence more general online help seeking behavior and use. However, I struggle with how to tease out or make the jump from quality assessments of specific platforms (in this case Good Doctor and others) and general online health seeking behavior when the measures used are specific to each platform. I also think the fact that participants were recruited for this study only if they were already using the platforms is problematic in regards to the conclusions that are being made. I am left wondering if there are more general measures of how an individual perceives the general quality of online health information, not specific to certain platforms, would be more applicable to this research.

Beyond the theoretical concerns I have, I have concerns about a) the lack of controls used in the model and b) the conclusions that were made based on the analysis. First, I see no mention of controls or if/how individual factors factored into the models, specifically gender, and if any controls were in place for whether the person had an existing previously diagnosed mental health condition. Second, the authors state "the results show that when college students choose more online health information seeking behavior, they can effectively understand their own mental health status and reduce anxiety, suspicion, and other negative emotions, thereby improving their mental health" (p.15). It's not clear that the participants were seeking information about their mental health when using these platforms. Second, while relationships between these variables were established, the causal mechanism is still under some scrutiny, especially without proper statistical controls. Causality can not be assumed, as this conclusion states, because there was no measure of mental health prior to online help-seeking that was then compared to mental health after online help-seeking.

In summary, I applaud the authors for their efforts in conducting and writing up this analysis. I am uncomfortable with the measures that were utilized and the conclusions that were then proposed. I think the theoretical model may have some merit, but needs more empirical testing and to tease out use of specific online platforms versus more general use of online platforms. I also think statistical controls, especially for individual level factors, need to be considered. 

Round 2

Reviewer 3 Report

Thank you for providing a detailed response to my earlier concerns. I find the paper much improved. The introduction/background is much easier to follow and more concise. The new clarification of online mental health information seeking behavior (OMHISB) helps the reader maintain focus on the topic of the article specific to mental health (as opposed to physical health) and provides a level of consistency that was missing before. The new paragraph on page 3 lines 19-28 is helpful, but should be checked for grammar and perhaps could be made more concise. I also appreciate the mention and specification of the controls that were used. Even though they were not shown to have a statistically significant effect, it is always helpful to ensure the reader knows what controls were used. Overall, the additional details on data collection (specifically the eligibility of the participants) and the improved description of the empirical analysis help me better see and comprehend the causal model in a way that was not as clear previously. Overall, this paper is much improved. 
